# Population Structure and Genetic Diversity of *Colletotrichum gloeosporioides* on *Citrus* in China

**Bei Liu [1], Xingxing Liang [2], Jinchao Kong [2], Chen Jiao [1], Hongye Li [1,\*] and Yunpeng Gai [2,\*]**

[1] Key Laboratory of Molecular Biology of Crop Pathogens and Insects, Institute of Biotechnology, Zhejiang University, Hangzhou 310058, China
[2] School of Grassland Science, Beijing Forestry University, Beijing 100083, China
[\*] Correspondence: hyli@zju.edu.cn (H.L.); gaiyunpeng@bjfu.edu.cn (Y.G.); Tel.: +86-13634190823 (H.L.); +86-18636836141 (Y.G.)

**Abstract:** To analyze the genetic structure and genetic diversity of *Colletotrichum gloeosporioides* as the dominant *Colletotrichum* species on Citrus, the Glyceraldehyde-3-phosphate dehydrogenase (GAPDH) genetic diversity, including 63 strains isolated and selected from 8 different sites and 5 different citrus species, was studied. A total of 19 GAPDH haplotypes were identified by genetic analysis, and the main haplotype (haplotype 5) was distributed in 28 isolates, mainly from *Citrus unchiu* Hort. ex Tanaka (WG) and *Citrus reticulata* Blanco cv. Succosa (BDZ) in Huangyan (HY), Linhai (LH), and Jiande (JD) of Zhejiang province, and *Mashui tangerine* (MSJ) in Mengshan of Guangxi province (GX). Using the genetic differentiation index, Fst revealed significant genetic differentiation in *C. gloeosporioides* populations between Jiangxi province (JXGZ) and GX, HY, LH, JD, and Chun'an (CA) of Zhejiang province, and also revealed slightly less genetic differentiation for *C. gloeosporioides* populations between HY, LH, JD, GX, Shaanxi province (SX), and Quzhou (QZ) of Zhejiang province. In addition, Fst revealed great genetic differentiation between the *C. gloeosporioides* populations obtained from MSJ and *Citrus paradise* Macf (PTY), and also revealed weak genetic differentiation between the *C. gloeosporioides* populations obtained from *Citrus sinensis* Osbeck (QC), WG, and BDZ. The AMOVA test showed that the levels of genetic differentiation for *C. gloeosporioides* were 19% and 81% among and within geographic populations, respectively. It also showed that *C. gloeosporioides* had levels of genetic differentiation among and within host populations of 12% and 88%, respectively. The Mantel test showed that the genetic distance was not linearly correlated with geographical distance and the haplotype phylogenetic analysis showed that *C. gloeosporioides* from different regions and hosts were scattered in the phylogenetic tree, implying that the genetic differentiation was independent of host variety and geographical origin. We speculated that genetic differentiation may be mainly due to gene mutation, gene recombination, or gene migration within native populations and has nothing to do with natural selection triggered by geography or host variety.

**Keywords:** *Colletotrichum gloeosporioides*; genetic structure; GAPDH sequence; haplotype

## 1. Introduction

Citrus is one of the most extensively produced fruit tree crops, and is widely distributed in tropical and subtropical regions in the world. The global citrus industry has developed rapidly, with a worldwide planting area of approximately 9.28 million hectares and an annual production of 158.50 million tons (Food and Agriculture Organization of the United Nations). China, Brazil, the United States, Mexico, and Spain are the five major citrus-producing countries in the world, accounting for more than 60% of the world's total production (http://www.fao.org/economic/est-commodities/citrus, accessed on 1 May 2020). In recent years, the world's citrus production has gradually shifted to advantageous regions, from the Americas to Asia and some African countries. The history of citrus cultivation in China dates back over 4000 years, making it one of the first citrus-growing

countries in the world [1]. At present, China is the world's largest citrus-producing country in the world, with a citrus planting area of 2.83 million hectares and an output of 55.95 million tons in 2020. Citrus is mainly produced in southern China, including Zhejiang, Fujian, Hunan, Sichuan, Guangxi, Hubei, Guangdong, Jiangxi, Chongqing, and Taiwan, followed by Shanghai, Guizhou, Yunnan, Jiangsu, Shaanxi, Henan, Hainan, Anhui, and Gansu Province, China.

Citrus has a long history of selection and cultivation worldwide, contributing to the emergence and spread of citrus fungal diseases. Citrus anthracnose is an important fungal disease caused by *Colletotrichum* spp, which is widespread in citrus-producing areas in the world [2]. The *Colletotrichum* genus is widely distributed in tropical and subtropical regions, with saprophytic, endophytic, and parasitic lifestyles. As one of the most prominent phytopathogenic fungi in the world [3], the genus *Colletotrichum* has an extremely wide range of hosts, such as vegetables, fruits, ornamental plants, and others [4]. Among them, *Colletotrichum gloeosporioides* is the most closely related fungus to citrus, which is found in the major citrus-producing regions all over the world [5]. Current research also confirms that *C. gloeosporioides* became the dominant *Colletotrichum* species on citrus in China [6].

In genetics, the haplotype is a combination of alleles genetically co-inherited at distinct loci on the same chromosome. The rich haplotype diversity of a population indicates great genetic diversity and genetic resources. They have a direct impact on the evolutionary potential of pathogenic populations, and play a significant role in the process of pathogenic populations adapting to new environments through evolution. Pathogenic populations with high genetic diversity have more survival and evolutionary advantages and can adapt more quickly to the presence of novel host resistance genes, chemical pesticides, and other disease control technologies [7].

In this study, based on the GAPDH gene sequence analysis of the genetic structure of *C. gloeosporioides* populations in different citrus varieties and geographic regions, the haplotype was used to elucidate whether the host population and geographic population of *C. gloeosporioides* exhibited gene flow, genetic differentiation, and host specificity at the molecular level, in order to preliminarily understand the correlation between the genetic structure, geography, and host source of *C. gloeosporioides* on citrus and to provide a reference for further research on genetic variation in the genus *Colletotrichum*.

## 2. Materials and Methods

### 2.1. Collection of Fungal Strains

The molecular and morphological identification of the strains indicated that there were 63 *C. gloeosporioides* on 5 distinct citrus hosts in 8 different locales. In order to explore the genetic structure of *C. gloeosporioides*, 63 strains were separated based on their respective host or geographic origins, including HY2~7, HY12, HY14, HY18, HY19, LH1~4, LH6, LH8~14, JD1~8, GX1~9, SX9~12, QZ1~7, QZ11~15, CA1~5, JZGZ1~3 or QC1~4, WG1~33, BDZ1~12, MSJ1~9, and PTY1~5. Detailed sample collecting information is listed in Table 1.

### 2.2. DNA Extraction, PCR Amplification, and Sequencing

The DNA of the *C. gloeosporioides* strain was extracted using the CTAB method [8]. The primer pairs GDF (5′-GCCGTCAACGACCCCTTCATTGA-3′) and GDR (5′-GGGTGGA GTCGTACTTGAGCATGT-3′) were used to amplify the partial GAPDH gene. The PCR reaction system (25 μL) included 1 μL of DNA, 1 μL each of upstream and downstream primers, 12.5 μL of 2 × PCR Master (including dye), and ddH$_2$O to 25 μL. The PCR reaction conditions included pre-denaturation at 94 °C for 4 min, 34 cycles of denaturation at 94 °C for 45 s, annealing at 58 °C for 45 s, extension at 72 °C for 1 min, final extension at 72 °C for 10 min, and storage at 4 °C [6]. The PCR product electrophoresis detection involved 50 mL 0.5×TBE, 0.5 g agarose, and 5 μL Glodview (nucleic acid stain) on a 1% agarose gel for electrophoresis detection. After detection, the PCR products were sent to Hangzhou Qingke Technology Company for bidirectional sequencing.

**Table 1.** Information of *Colletotrichum gloeosporioides* strains collected from citrus in China.

| Strain | Site (Code) | Longitude, Latitude | Host (Code) | Genbank Accession of GAPDH | Hyplotype (Hn) | Collection Year |
|---|---|---|---|---|---|---|
| HY2 | Huangyan, Zhejiang (HY) | 121.17, 28.64 | *Citrus sinensis* Osbeck (QC) | MT449239 | H5 | 2018 |
| HY3 | Huangyan, Zhejiang (HY) | 121.17, 28.64 | *Citrus sinensis* Osbeck (QC) | MT449240 | H5 | 2018 |
| HY4 | Huangyan, Zhejiang (HY) | 121.17, 28.64 | *Citrus unchiu* Hort. ex Tanaka (WG) | MT449241 | H19 | 2018 |
| HY5 | Huangyan, Zhejiang (HY) | 121.17, 28.64 | *Citrus unchiu* Hort. ex Tanaka (WG) | MT449242 | H17 | 2018 |
| HY6 | Huangyan, Zhejiang (HY) | 121.17, 28.64 | *Citrus reticulata* Blanco cv. Succosa (BDZ) | MT449243 | H5 | 2018 |
| HY7 | Huangyan, Zhejiang (HY) | 121.17, 28.64 | *Citrus reticulata* Blanco cv. Succosa (BDZ) | MT449244 | H5 | 2018 |
| HY12 | Huangyan, Zhejiang (HY) | 121.17, 28.64 | *Citrus unchiu* Hort. ex Tanaka (WG) | MT449249 | H6 | 2018 |
| HY14 | Huangyan, Zhejiang (HY) | 121.17, 28.64 | *Citrus unchiu* Hort. ex Tanaka (WG) | MT449251 | H1 | 2018 |
| HY18 | Huangyan, Zhejiang (HY) | 121.17, 28.64 | *Citrus reticulata* Blanco cv. Succosa (BDZ) | MT449255 | H1 | 2018 |
| HY19 | Huangyan, Zhejiang (HY) | 121.17, 28.64 | *Citrus reticulata* Blanco cv. Succosa (BDZ) | MT449256 | H5 | 2018 |
| LH1 | Linhai, Zhejiang (LH) | 121.29, 28.76 | *Citrus unchiu* Hort. ex Tanaka (WG) | MT449257 | H1 | 2018 |
| LH2 | Linhai, Zhejiang (LH) | 121.29, 28.76 | *Citrus unchiu* Hort. ex Tanaka (WG) | MT449258 | H5 | 2018 |
| LH3 | Linhai, Zhejiang (LH) | 121.29, 28.76 | *Citrus reticulata* Blanco cv. Succosa (BDZ) | MT449259 | H5 | 2018 |
| LH4 | Linhai, Zhejiang (LH) | 121.29, 28.76 | *Citrus reticulata* Blanco cv. Succosa (BDZ) | MT449260 | H5 | 2018 |
| LH6 | Linhai, Zhejiang (LH) | 121.29, 28.76 | *Citrus reticulata* Blanco cv. Succosa (BDZ) | MT449262 | H5 | 2018 |
| LH8 | Linhai, Zhejiang (LH) | 121.29, 28.76 | *Citrus reticulata* Blanco cv. Succosa (BDZ) | MT449264 | H6 | 2018 |
| LH9 | Linhai, Zhejiang (LH) | 121.29, 28.76 | *Citrus unchiu* Hort. ex Tanaka (WG) | MT449265 | H7 | 2018 |
| LH10 | Linhai, Zhejiang (LH) | 121.29, 28.76 | *Citrus reticulata* Blanco cv. Succosa (BDZ) | MT449266 | H5 | 2018 |
| LH11 | Linhai, Zhejiang (LH) | 121.29, 28.76 | *Citrus reticulata* Blanco cv. Succosa (BDZ) | MT449267 | H5 | 2018 |
| LH12 | Linhai, Zhejiang (LH) | 121.29, 28.76 | *Citrus reticulata* Blanco cv. Succosa (BDZ) | MT449268 | H5 | 2018 |
| LH13 | Linhai, Zhejiang (LH) | 121.29, 28.76 | *Citrus reticulata* Blanco cv. Succosa (BDZ) | MT449269 | H5 | 2018 |
| LH14 | Linhai, Zhejiang (LH) | 121.29, 28.76 | *Citrus unchiu* Hort. ex Tanaka (WG) | KC293710 | H5 | 2018 |
| JD1 | Jiande, Zhejiang (JD) | 119.56, 29.54 | *Citrus unchiu* Hort. ex Tanaka (WG) | MT449270 | H5 | 2018 |
| JD2 | Jiande, Zhejiang (JD) | 119.56, 29.54 | *Citrus unchiu* Hort. ex Tanaka (WG) | MT449271 | H5 | 2018 |
| JD3 | Jiande, Zhejiang (JD) | 119.56, 29.54 | *Citrus unchiu* Hort. ex Tanaka (WG) | MT449272 | H6 | 2018 |
| JD4 | Jiande, Zhejiang (JD) | 119.56, 29.54 | *Citrus unchiu* Hort. ex Tanaka (WG) | MT449273 | H5 | 2018 |
| JD5 | Jiande, Zhejiang (JD) | 119.56, 29.54 | *Citrus unchiu* Hort. ex Tanaka (WG) | MT449274 | H5 | 2018 |
| JD6 | Jiande, Zhejiang (JD) | 119.56, 29.54 | *Citrus unchiu* Hort. ex Tanaka (WG) | MT449275 | H6 | 2018 |
| JD7 | Jiande, Zhejiang (JD) | 119.56, 29.54 | *Citrus unchiu* Hort. ex Tanaka (WG) | MT449276 | H5 | 2018 |
| JD8 | Jiande, Zhejiang (JD) | 119.56, 29.54 | *Citrus unchiu* Hort. ex Tanaka (WG) | MT449277 | H11 | 2018 |
| GX1 | Mengshan, Guangxi (GX) | 110.53, 24.2 | Mashui tangerine (MSJ) | MT449278 | H5 | 2018 |
| GX2 | Mengshan, Guangxi (GX) | 110.53, 24.2 | Mashui tangerine (MSJ) | MT449279 | H5 | 2018 |
| GX3 | Mengshan, Guangxi (GX) | 110.53, 24.2 | Mashui tangerine (MSJ) | MT449280 | H5 | 2018 |
| GX4 | Mengshan, Guangxi (GX) | 110.53, 24.2 | Mashui tangerine (MSJ) | MT449281 | H5 | 2018 |
| GX5 | Mengshan, Guangxi (GX) | 110.53, 24.2 | Mashui tangerine (MSJ) | MT449282 | H5 | 2018 |
| GX6 | Mengshan, Guangxi (GX) | 110.53, 24.2 | Mashui tangerine (MSJ) | MT449283 | H5 | 2018 |
| GX7 | Mengshan, Guangxi (GX) | 110.53, 24.2 | Mashui tangerine (MSJ) | MT449284 | H1 | 2018 |
| GX8 | Mengshan, Guangxi (GX) | 110.53, 24.2 | Mashui tangerine (MSJ) | MT449285 | H5 | 2018 |
| GX9 | Mengshan, Guangxi (GX) | 110.53, 24.2 | Mashui tangerine (MSJ) | MT449286 | H1 | 2018 |
| SX9 | Chenggu, Shanxi (SX) | 109.03, 34.31 | *Citrus unchiu* Hort. ex Tanaka (WG) | KC293717 | H9 | 2012 |
| SX10 | Chenggu, Shanxi (SX) | 109.03, 34.31 | *Citrus unchiu* Hort. ex Tanaka (WG) | KC293718 | H2 | 2012 |
| SX11 | Chenggu, Shanxi (SX) | 109.03, 34.31 | *Citrus unchiu* Hort. ex Tanaka (WG) | KC293719 | H5 | 2011 |
| SX12 | Chenggu, Shanxi (SX) | 109.03, 34.31 | *Citrus unchiu* Hort. ex Tanaka (WG) | KC293720 | H3 | 2011 |
| QZ1 | Quzhou, Zhejiang (QZ) | 118.88, 28.98 | *Citrus unchiu* Hort. ex Tanaka (WG) | MT449295 | H10 | 2019 |
| QZ2 | Quzhou, Zhejiang (QZ) | 118.88, 28.98 | *Citrus unchiu* Hort. ex Tanaka (WG) | MT449296 | H11 | 2019 |
| QZ3 | Quzhou, Zhejiang (QZ) | 118.88, 28.98 | *Citrus unchiu* Hort. ex Tanaka (WG) | MT449297 | H11 | 2019 |
| QZ4 | Quzhou, Zhejiang (QZ) | 118.88, 28.98 | *Citrus unchiu* Hort. ex Tanaka (WG) | MT449298 | H15 | 2019 |
| QZ5 | Quzhou, Zhejiang (QZ) | 118.88, 28.98 | *Citrus unchiu* Hort. ex Tanaka (WG) | MT449299 | H11 | 2019 |
| QZ6 | Quzhou, Zhejiang (QZ) | 118.88, 28.98 | *Citrus unchiu* Hort. ex Tanaka (WG) | MT449300 | H14 | 2019 |
| QZ7 | Quzhou, Zhejiang (QZ) | 118.88, 28.98 | *Citrus unchiu* Hort. ex Tanaka (WG) | MT449301 | H15 | 2019 |
| QZ11 | Quzhou, Zhejiang (QZ) | 118.88, 28.98 | *Citrus paradise* Macf (PTY) | MT449305 | H15 | 2019 |
| QZ12 | Quzhou, Zhejiang (QZ) | 118.88, 28.98 | *Citrus paradise* Macf (PTY) | MT449306 | H11 | 2019 |
| QZ13 | Quzhou, Zhejiang (QZ) | 118.88, 28.98 | *Citrus paradise* Macf (PTY) | MT449307 | H18 | 2019 |
| QZ14 | Quzhou, Zhejiang (QZ) | 118.88, 28.98 | *Citrus paradise* Macf (PTY) | MT449308 | H13 | 2019 |
| QZ15 | Quzhou, Zhejiang (QZ) | 118.88, 28.98 | *Citrus paradise* Macf (PTY) | MT449309 | H8 | 2019 |
| CA1 | Chun'an, Zhejiang (CA) | 118.56, 29.4 | *Citrus unchiu* Hort. ex Tanaka (WG) | MT449310 | H11 | 2019 |
| CA2 | Chun'an, Zhejiang (CA) | 118.56, 29.4 | *Citrus unchiu* Hort. ex Tanaka (WG) | MT449311 | H15 | 2019 |
| CA3 | Chun'an, Zhejiang (CA) | 118.56, 29.4 | *Citrus unchiu* Hort. ex Tanaka (WG) | MT449312 | H12 | 2019 |
| CA4 | Chun'an, Zhejiang (CA) | 118.56, 29.4 | *Citrus unchiu* Hort. ex Tanaka (WG) | MT449313 | H16 | 2019 |
| CA5 | Chun'an, Zhejiang (CA) | 118.56, 29.4 | *Citrus unchiu* Hort. ex Tanaka (WG) | MT449314 | H8 | 2019 |
| JXGZ1 | Ganzhou, Jiangxi (JXGZ) | 114.9, 25.8 | *Citrus sinensis* Osbeck (QC) | KC293706 | H4 | 2011 |
| JXGZ2 | Ganzhou, Jiangxi (JXGZ) | 114.9, 25.8 | *Citrus sinensis* Osbeck (QC) | KC293707 | H9 | 2011 |
| JXGZ3 | Ganzhou, Jiangxi (JXGZ) | 114.9, 25.8 | *Citrus unchiu* Hort. ex Tanaka (WG) | KC293708 | H4 | 2011 |

Note: Hn represents n hyplotype.

### 2.3. Data Analysis

The GAPDH gene sequences of 63 strains were aligned using the MEGA6.0 program, and the base composition and variation site information were tallied after shearing and manual correction [9]. Haplotype diversity is the frequency at which two haplotypes are randomly selected within a population. Therefore, the more haplotype varieties, the greater the genetic diversities and genetic resources in a population. In this study, the GAPDH locus was only investigated, and haplotype diversity represents the diversity of allele combinations on the GAPDH gene. The software GenALEx6 was used to examine the haplotype in addition to the haplotype diversity and molecular variation (AMOVA) of *C. gloeosporioides* populations [10,11]. Using TCS1.21, a network relationship map of sample haplotypes can be created [12]. The genetic differentiation index (Fst) quantifies the level of population differentiation and is typically calculated using genetic polymorphism data. As a special case of the Wright's F statistic, Fst is one of the most commonly used statistics in population genetics and can be analyzed using DNAsp v5.0 [13]. The relationship between genetic difference and geographical distance was explored using the Mantel test [14]. Using PCoA (Principal Coordinate Analysis) to analyze the similarities or differences in genetic data, the Fst information can be visualized in order to observe the differences between individuals or populations. DNAsp5.0 was used to analyze haplotype mismatches in all the geographic population sequences in order to determine whether the population had experienced expansion or continued growth in the past. The software MEGA6.0 was used to create phylogenetic trees of distinct haplotypes [9].

## 3. Results

### 3.1. GAPDH Sequence Haplotype Diversity

The GAPDH gene sequence fragment of the *C. gloeosporioides* population consists of 247 bp, 22 polymorphic sites, and 19 haplotypes (H). As seen in Table 2, the majority of these strains had the haplotype H5 (Hyplotype5), accounting for 44% of the total samples. QZ and WG have the largest number of haplotypes among geographic and host populations, containing 7 and 17 haplotypes, respectively. GX, JXGZ, and MSJ have the smallest number of haplotypes among geographical and host populations, all containing two haplotypes. Some geographic populations and host populations have their own unique haplotypes. H2 and H3 are exclusive to WG grown in SX, whereas H13 and H18 are exclusive to PTY grown in QZ. There is genetic differentiation of citrus varieties in the region of this study. The haplotype diversity of each geographic and host population of *C. gloeosporioides* ranged from 0.389 to 1.000 and from 0.318 to 1.000, respectively. SX, CA, and PTY populations have the highest haplotype diversity with values of 1, implying that each strain in the population has its own unique haplotype. However, haplotype diversity was lowest in the GX and BDZ populations at 0.389 and 0.318, respectively.

**Table 2.** Summary of AMOVA tests within and among the geographic populations and host populations of *Colletotrichum gloeosporioides* from *Citrus* in China.

| Source | df | SS | MS | Est. Var. | % | Stat | Value | Prob (P) |
|--------|----|----|-----|-----------|-----|------|-------|----------|
| geographic populations | | | | | | | | |
| Among Pops | 7 | 6.415 | 0.916 | 0.077 | 19% | PhiPT | 0.190 | 0.001 |
| Within Pops | 55 | 17.997 | 0.327 | 0.327 | 81% | | | |
| Total | 62 | 24.413 | | 0.404 | 100% | | | |
| host populations | | | | | | | | |
| Among Pops | 4 | 3.403 | 0.851 | 0.047 | 12% | PhiPT | 0.115 | 0.002 |
| Within Pops | 58 | 21.010 | 0.362 | 0.362 | 88% | | | |
| Total | 62 | 24.413 | | 0.409 | 100% | | | |

### 3.2. Haplotype Network Analysis

The haplotype network structure was plotted using the TCS1.21 program, which showed that *C. gloeosporioides* produced large genetic differentiation and that the haplotype

did not form their own branches according to sample location or citrus variety, but presented an uncommon stellate divergence where H5 (strain HY2) was the original haplotype at the center of radiation. From the analysis of geographical sources, it was widely distributed in HY (5), LH (9), JD (6), and GX (7). From the host source analysis, it was mainly parasitized in WG (9), BDZ (10), and MSJ (7). This may be related to the long history of citrus cultivation in our country in Huangyan, Linhai, Jiande in Zhejiang province, and Guangxi province, resulting in the more widespread existence of primitive haplotypes there. Other haplotypes are generated on the basis of this haplotype through one or several mutations. Specially, H11 (strain JD8) finally becomes a subcenter during gradual variation and fixation (Figure 1).

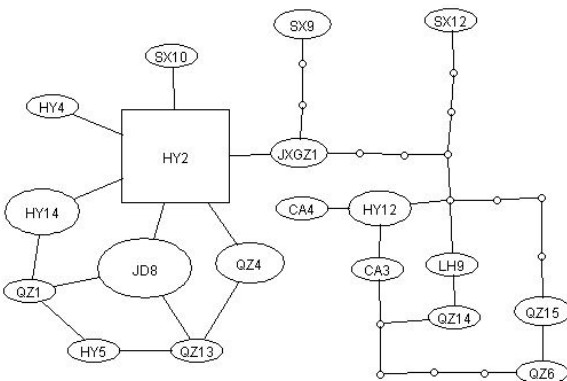

**Figure 1.** Parsimony network of hyplotypes of *Colletotrichum gloeosporioides* from *Citrus* in China. Each mutation is shown as a line between every two haplotypes. The small dots represent unsampled haplotypes.

### 3.3. Genetic Differentiation Analysis

In Table 3, the AMOVA study of eight geographic populations and five host populations showed that the population variation of *C. gloeosporioides* from citrus in China mainly occurred within the populations, accounting for 81% and 88% of the total variation, respectively. However, the genetic differences among the populations were small, accounting for 19% and 12%, suggesting that the gene flow among the *C. gloeosporioides* populations was frequent, and that the genes might be exchanged through the transport of citrus propagation materials in different regions. The variation within the population was significantly greater than 0, indicating that the genetic variation within the geographic and host populations of *C. gloeosporioides* was significant ($p < 0.05$).

The Fst values in Table 4 show that the genetic differentiation values between the geographic population CA and geographic populations HY, LH, JD, and GX vary from 0.144 to 0.368, indicating that the genetic divergence between these populations is significant and there are genetic exchange barriers. The genetic differentiation value between JXGZ and HY, LH, JD, GX, and CA was greater than 0.25, indicating that the genetic differentiation is very strong, whereas the genetic differentiation values between HY, LH, JD, and GX, SX, and QZ were small, showing that there is gene exchange with weak genetic differentiation. On the other hand, genetic differentiation values between the host populations MSJ and QC and WG, and host populations PTY and QC, WG, and BDZ range from 0.051 to 0.137, indicating that these populations were moderately differentiated. The Fst value between MSJ and PTY was 0.164, indicating relatively higher genetic differentiation between these populations. The results for genetic differentiation among QC, WG, and BDZ were all less than 0.05, showing that the level of genetic differentiation in these populations was low and gene exchange was frequent. The Mantel test found that there was no significant linear relationship between geographical distance and genetic distance between the populations ($p = 0.17 > 0.05$).

**Table 3.** Pairwise Fst values between the geographical populations and host populations of *Colletotrichum gloeosporioides* from *Citrus* in China.

| Geographical Populations | HY | LH | JD | GX | SX | QZ | CA | JXGZ | Host Populations | QC | WG | BDZ | MSJ | PTY |
|---|---|---|---|---|---|---|---|---|---|---|---|---|---|---|
| HY | 0.000 | | | | | | | | QC | 0.000 | | | | |
| LH | 0.052 | 0.000 | | | | | | | WG | 0.047 | 0.000 | | | |
| JD | 0.056 | 0.096 | 0.000 | | | | | | BDZ | 0.048 | 0.029 | 0.000 | | |
| GX | 0.039 | 0.052 | 0.028 | 0.000 | | | | | MSJ | 0.132 | 0.137 | 0.012 | 0.000 | |
| SX | 0.005 | 0.051 | 0.046 | 0.065 | 0.000 | | | | PTY | 0.069 | 0.075 | 0.051 | 0.164 | 0.000 |
| QZ | 0.020 | 0.030 | 0.010 | 0.123 | 0.040 | 0.000 | | | - | - | - | - | - | - |
| CA | 0.223 | 0.144 | 0.183 | 0.368 | 0.049 | 0.027 | 0.000 | | - | - | - | - | - | - |
| JXGZ | 0.321 | 0.270 | 0.300 | 0.462 | 0.102 | 0.220 | 0.280 | 0.000 | - | - | - | - | - | - |

Note: the genetic differentiation index (Fst): 0~0.05 is very weak, marked by gray boxes; 0.05~0.15 is moderate, marked by green boxes; 0.15~0.25 is relatively high, marked by yellow boxes; greater than 0.25 indicates great differentiation and is marked by pink boxes in the figure.

**Table 4.** Genetic diversity of the *Colletotrichum gloeosporioides* population from *Citrus* in China.

| Hn (Number of Strains) | Geographical Origin | | | | | | | | Host Origin | | | | |
|---|---|---|---|---|---|---|---|---|---|---|---|---|---|
| | HY (n) | LH (n) | JD (n) | GX (n) | SX (n) | QZ (n) | CA (n) | JXGZ (n) | QC (n) | WG (n) | BDZ (n) | MSJ (n) | PTY (n) |
| | H1 (2) H5 (5) H6 (1) H17 (1) H19 (1) | H1 (1) H5 (9) H6 (1) H7 (1) | H5 (6) H6 (1) H11 (1) | H1 (2) H5 (7) | H2 (1) H3 (1) H5 (1) H9 (1) | H8 (1) H10 (1) H11 (4) H13 (1) H14 (1) H15 (3) H18 (1) | H8 (1) H11 (1) H12 (1) H15 (1) H16 (1) | H4 (2) H9 (1) | H4 (1) H5 (2) H9 (1) | H1 (2) H2 (1) H3 (1) H4 (1) H5 (9) H6 (2) H7 (1) H8 (1) H9 (1) H10 (1) H11 (5) H12 (1) H14 (1) H15 (3) H16 (1) H17 (1) H19 (1) | H1 (1) H5 (10) H6 (1) | H1 (2) H5 (7) | H8 (1) H11 (1) H13 (1) H15 (1) H18 (1) |
| Total number of strains | 10 | 12 | 8 | 9 | 4 | 12 | 5 | 3 | 4 | 33 | 12 | 9 | 5 |
| Haploid diversity (h) | 0.680 | 0.417 | 0.406 | 0.346 | 0.750 | 0.792 | 0.800 | 0.444 | 0.625 | 0.876 | 0.292 | 0.346 | 0.800 |
| Unbiased diversity(uh) | 0.756 | 0.455 | 0.464 | 0.389 | 1.000 | 0.864 | 1.000 | 0.667 | 0.833 | 0.903 | 0.318 | 0.389 | 1.000 |

Note: n = number of strains; uh = Unbiased Diversity = $(N/(N-1)) \times h$; HY, LH, JD, GX, SX, QZ, CA, and JXGZ represent the geographic population of Huangyan, Linhai, Jiande, Guangxi, Shanxi, Quzhou, Chun'an, and Ganzhou, Jiangxi province; QC, WG, BDZ, MSJ, and PTY represent the host population of *Citrus sinensis* Osbeck, *Citrus unchiu* Hort. ex Tanaka, *Citrus reticulata* Blanco cv. Succosa, Mashui tangerine, and *Citrus paradise* Macf.

### 3.4. Population Expansion Analysis

The mismatch distribution curve will indicate a unimodal Poisson distribution when the population has expanded or grown continuously throughout history, and a multimodal distribution when the population size has remained stable. After employing a mismatch analysis of the GAPDH sequences from all the populations of *C. gloeosporioides*, the distribution curve has three peaks, indicating a multimodal curve (Figure 2) that does not conform to the unimodal curve pattern of population expansion. This suggests that the *C. gloeosporioides* populations may not have undergone large-scale population expansion.

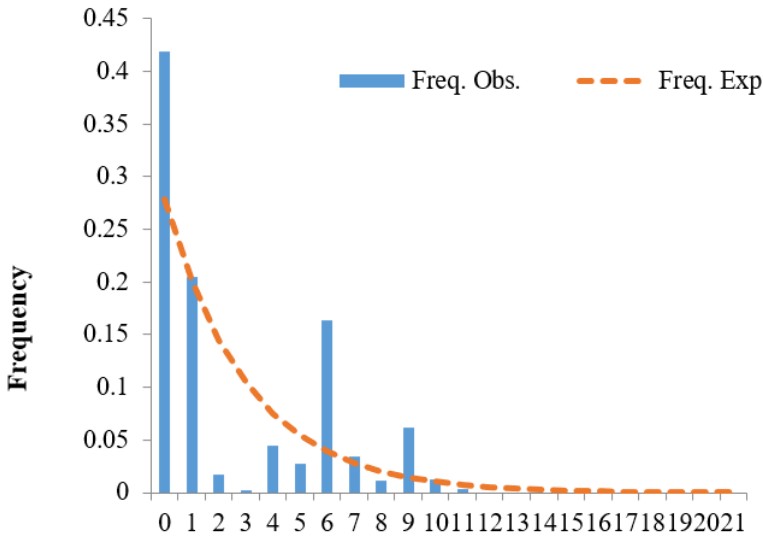

**Figure 2.** Mismatch distribution of the *Colletotrichum gloeosporioides* population from *Citrus* in China. Exp stands for expected value and Obs stands for observed value.

### 3.5. Phylogenetic Relationship Analysis Based on GAPDH Sequences

*Colletotrichum truncatum* GAPDH sequences from the GenBank database were selected as outgroups, and a phylogenetic tree with 19 distinct haplotype sequences was created, with each strain representing a haplotype. Figure 3 shows that all the haplotypes were mostly split into three main clades: clade 1 had 11 haplotypes with a 64% support rate, clade 2 had 3 haplotypes with a 66% support rate, and clade 3 had 2 haplotypes with a 60% support rate. In addition, the same clade includes strains of different geographical origins, for instance, eight strains with various geographical origins are gathered together in clade 1. The strains from the same host are distributed in different clades. For example, the strains HY4, LH9, and CA3 from *Citrus sinensis* Osbeck are distributed in clades 1, 2, and 3, respectively, indicating that the genetic variation of *C. gloeosporioides* is independent of the host and geography mainly due to individual genetic mutations, gene recombinations, or gene migration, and has nothing to do with the natural selection that may be caused by the geography or host species. The phylogenetic tree shows that the haplotypes for each region and host are dispersed, demonstrating that there is some efficient gene flow between the regions and the hosts in the *C. gloeosporioides* populations.

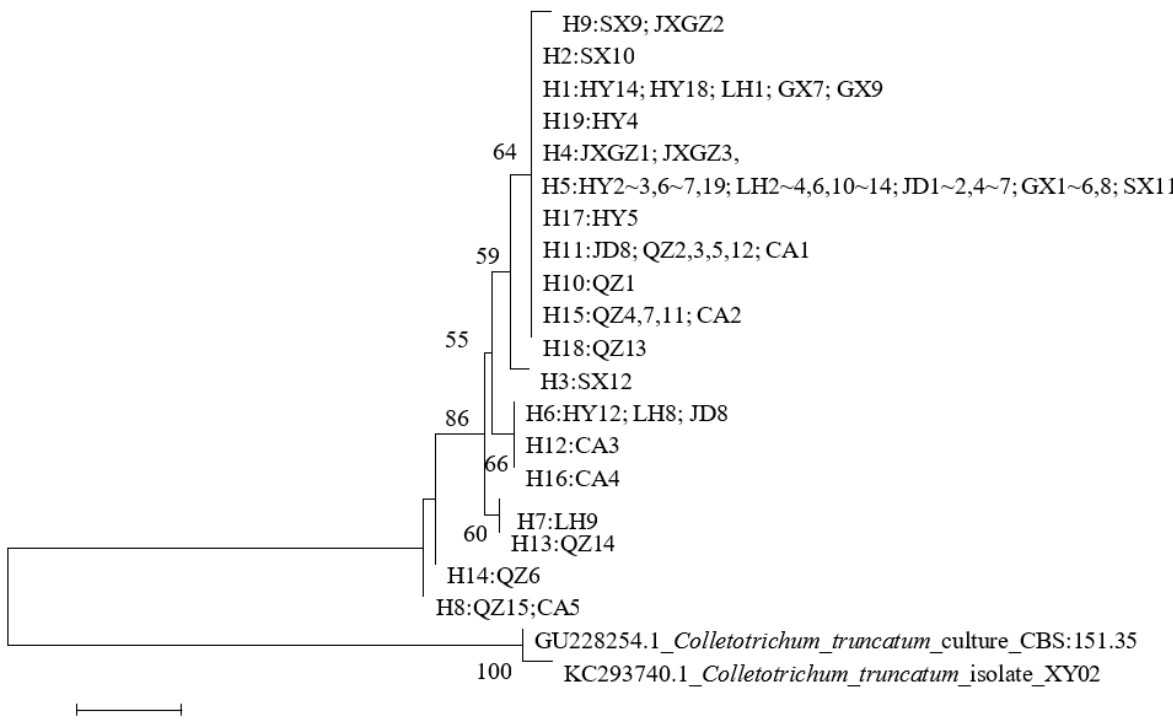

**Figure 3.** The neighbour-joining tree of the GAPDH sequences of *Colletotrichum gloeosporioides* and representative sequences from *Colletotrichum truncatum* as an outgroup.

## 4. Discussion

Significant haplotype diversity indicates abundant genetic variation and genetic resources in a population. The genetic diversity of pathogenic microbial populations has a direct impact on their evolutionary potential and plays an important role in the process of adaptation to new environments. In this study, the mean haplotype diversity of the *C. gloeosporioides* population from citrus in China was 0.7, that is, the probability of two strains with distinct haplotypes in the population was 70%. This indicates that the *C. gloeosporioides* population from citrus in China has a high genetic diversity and may swiftly respond to novel host resistance genes, chemical pesticides, and other disease control strategies.

Through genetic analysis of the differences among the ITS sequences of 126 *Camellia oleifera* from 12 different regions in 6 provinces of China, it was concluded that the *C. gloeosporioides* populations from Changde and Tianjiling in Hunan province possessed high genetic diversity. This is because Changde and Tianjiling are two long-term planting areas with more than 50 years of cultivation, which have richer genetic polymorphisms than other new planting areas [15]. This is consistent with the extremely rich genetic diversity of the *C. gloeosporioides* populations of HY, LH, JD, and GX, which also have a long history of citrus cultivation in the research, of which H5 is distributed in all the populations except QZ, CA, JXGZ, and the host population PTY. This indicates that *C. gloeosporioides* has the potential to transmit gene flow over long distances, and the asexual reproduction pattern of hyphae transmission between different regions leads to the existence of the same genotypes in different geographical groups, which mutate and further evolve into other haplotypes.

Previous research has found that the genetic variation of *C. gloeosporioides* is strong and has no relationship with the geographical origin, host species, or pathogenic tissue [6]. A previous study has shown that there was no connection between geographic distance and genetic variation [16]. This is consistent with the results of this study, where *C. gloeosporioides* exhibits significant genetic differentiation within geographic populations, and there is no meaningful association between geographic distance and genetic distance in a population.

The number of strains in each site varies greatly despite the fact that the materials used in this survey have a wide geographical source. For example, there are only three

strain samples in Ganzhou, Jiangxi province, and five strain samples in Chun'an, Zhejiang province, which cannot objectively reflect the number and diversity of haplotypes in these places. Nevertheless, Huangyan, and Quzhou have 10 and 12 strains, respectively, and the number of haplotypes in these areas demonstrated considerable diversity. The results of the many samples that have already been collected also fully show that *C. gloeosporioides* on *Citrus* from China has a very high haplotype diversity, indicating that the fungus has a strong potential for environmental adaptation and that it is easy to further expand its distribution range. This partly illustrates the possibility for serious widespread citrus diseases by *C. gloeosporioides* in China. In future studies, we can add more individuals and additional molecular markers, collect more accurate genetic data, and fully understand the genetic structure and diversity of the geographic and host populations of *C. gloeosporioides*, which can provide a more precise and reliable foundation for developing an overall plan of action for the disease management of *C. gloeosporioides*.

**Author Contributions:** Conceptualization, B.L., H.L. and Y.G.; methodology, B.L.; software, B.L.; validation, B.L. and Y.G.; formal analysis, B.L.; investigation, B.L.; resources, B.L.; data curation, B.L.; writing—original draft preparation, B.L., X.L. and J.K.; writing—review and editing, B.L., C.J. and Y.G.; visualization, B.L.; supervision, Y.G. and H.L.; project administration, C.J., Y.G. and H.L.; funding acquisition, Y.G. and H.L. All authors have read and agreed to the published version of the manuscript.

**Funding:** This study was supported by the National Natural Science Foundation of China (No.: 32001847) and Chinese Modern Agricultural Technology Systems (CARS-26).

**Data Availability Statement:** Data supporting reported results can be found GenBank database (MT449239-MT449314).

**Acknowledgments:** We would like to thank Xiaoe Xiao, Tao Xiong, Haijie Ma, Yating Zeng, Yanan Chen, and Huilan Fu (Institute of Biotechnology, Zhejiang University, China) for their essential support and advice.

**Conflicts of Interest:** The authors declare no conflict of interest.

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
