# Peer review of "Population Structure and Genetic Diversity of Colletotrichum gloeosporioides on Citrus in China"

_agronomy, doi:10.3390/agronomy13010184_

Round 1

Reviewer 1 Report

General comments The peer-reviewed manuscript entitled “Population structure and Genetic diversity of Colletotrichum gloeosporioides as dominant population on Citrus in China” contains relevant information on the title topic. The article deserves to be published, after the indicated corrections. The subject is interesting and it is advisable to continue research, which will be helpful in developing a strategy of control diseases caused by Colletotrichum gloeosporioides. The research methods and statistical tools used are correct and the conclusions are justified. In the Materials and Methods section, additional information on sampling is indicated. From which parts of Citrus plants (twig, leaves, fruit) were the mushroom isolates obtained? Did they show visual symptoms of damage?   Detailed comments Stylistic and typing corrections are recommended: I would suggest changing the title of the article to the following: “Population structure and genetic diversity of Colletotrichum gloeosporioides on (found on) Citrus L. sp. in China” or “Population structure and genetic diversity of Colletotrichum gloeosporioides from Citrus L. plants in China”. A similar title was in the review request. It seems to be better than the title in the manuscript.   l. 46. Colletotrichum genus is widely”…. Better to write: “Mentioned genus is…”. Then we avoid that the same word is at the end and the beginning of the following sentences.

l.: 13, 34, 62. GAPDH (Glyceraldehyde- 3-phosphate dehydrogenase) the full name should be entered when using the abbreviation for the first time. Meanwhile, we find the explanation of the abbreviation only on line 80.  

l. 181 Citrus sinensis should be in italics

l. 189-192. “Significant haplotype diversity indicates abundant genetic variation and genetic resources in a population, and the genetic diversity of pathogenic microbial populations has a direct impact on their evolutionary potential and plays an important role in the process of adapting to evolutionary adaptation to new environments”. Better should be: “Significant haplotype diversity indicates abundant genetic variation and genetic resources in a population. The genetic diversity of pathogenic microbial populations has a direct impact on their evolutionary potential and plays an important role in the process of adaptation to new environments”.

Date of manuscript received: 9 November 2022

Date of this review: 14 November 2022

Author Response

First of all, thank you very much for taking your time to review the manuscript. I believe your consideration and correction will add more value to the paper. On behalf of all the authors, I really appreciate the feedback from the Reviewer.

The following is my suggestions and comments:

A: Thanks for your advice. I have changed it.

  1. 46. „Colletotrichum genus is widely”…. Better to write: “Mentioned genus is…”. Then we avoid that the same word is at the end and the beginning of the following sentences.

A: Thanks for your advice. I have corrected it.

l.: 13, 34, 62. GAPDH (Glyceraldehyde- 3-phosphate dehydrogenase) the full name should be entered when using the abbreviation for the first time. Meanwhile, we find the explanation of the abbreviation only on line 80.  

A: Thanks for your advice. I have corrected it.

  1. 181 Citrus sinensis should be in italics

A: Thanks for your advice. I have corrected it.

  1. 189-192. “Significant haplotype diversity indicates abundant genetic variation and genetic resources in a population, and the genetic diversity of pathogenic microbial populations has a direct impact on their evolutionary potential and plays an important role in the process of adapting to evolutionary adaptation to new environments”. Better should be: “Significant haplotype diversity indicates abundant genetic variation and genetic resources in a population. The genetic diversity of pathogenic microbial populations has a direct impact on their evolutionary potential and plays an important role in the process of adaptation to new environments”.

A: Thanks for your advice. I have changed it.

Reviewer 2 Report

the manuscript was written well, the topic also is interesting for the readers,

I have a few points that should be corrected before publishing

2. Materials and Methods
2.1. Fungal Strains collecting
the authors should indicate here the location where they had samples collected (  geographic origins)

2.2 DNA extraction, PCR amplification, and sequencing,
...the authors should summarize this method (this method was written in more detail)
the authors should write a conclusion after the discussion section

Author Response

First of all, thank you very much for taking your time to review the manuscript. I believe your consideration and correction will add more value to the paper. On behalf of all the authors, I really appreciate the feedback from the Reviewer.

Reviewer2:

the manuscript was written well, the topic also is interesting for the readers,

Thanks so much!

I have a few points that should be corrected before publishing

  1. Materials and Methods

2.1. Fungal Strains collecting

the authors should indicate here the location where they had samples collected ( geographic origins)

A: Thanks for your advice. I show the information in Table 1.

2.2 DNA extraction, PCR amplification, and sequencing,

...the authors should summarize this method (this method was written in more detail)

the authors should write a conclusion after the discussion section

A: Thanks for your advice. The conclusion I want to draw is a few sentences at the end of each paragraph in the discussion section.

Reviewer 3 Report

The manuscript by Bei Liu et al. entitled “Population structure and Genetic diversity of Colletotrichum gloeosporioides as dominant population on Citrus in China” reports that genetic structure of C. gloeosporioides populations hosted in different citrus varieties and distributed in various regions based on GAPDH sequence analysis. The authors then claim to elucidate whether the host population and geographic population of C. gloeosporioides existed gene flow, genetic differentiation, and host specificity at the molecular level, in order to preliminarily understand the correlation between the genetic structure, geography and host source of C. gloeosporioides on citrus. The work is a technically sound piece of research, but many unsure results presented. It requires major revision before its acceptance for publication.

1.      The authors should rewrite the text with helps from a professional or a native English speaker peer. For instance, lane 42 of Introduction ‘from the Chinese mainland’ should be ‘from Mainland, China’.  

2.      Detailed information of the M&M, for example, the primer pair sequences.

3.      The PCR products sequencing results do not representative a single molecule sequence, the authors need provide chromatograms to verify the nucleotide variation (or SNP) results; or sequencing cloned PCR products.

4.      The authors need clarify the source of the PCR reaction system, and let readers know if the DNA polymerase used in this research is high fidelity or not.

5.      The authors need provide the images of single colony morphology of the representative isolates. 

Author Response

First of all, thank you very much for taking your time to review the manuscript. I believe your consideration and correction will add more value to the paper. On behalf of all the authors, I really appreciate the feedback from the Reviewer.

  1. The authors should rewrite the text with helps from a professional or a native English speaker peer. For instance, lane 42 of Introduction ‘from the Chinese mainland’ should be ‘from Mainland, China’.  

A: Thank you I have corrected.

  1. Detailed information of the M&M, for example, the primer pair sequences.

A: Thank you I have added.

  1. The PCR products sequencing results do not representative a single molecule sequence, the authors need provide chromatograms to verify the nucleotide variation (or SNP) results; or sequencing cloned PCR products.

A: I have identified them from morphological and molecular aspect in my Master's thesis. They all belong to the species of C. gloeosporioides.

  1. The authors need clarify the source of the PCR reaction system, and let readers know if the DNA polymerase used in this research is high fidelity or not.

A: To make the article concise , I did not show DNA polymerase in details but directly show DNA polymerase of C. gloeosporioides from different host and geographical regions. We also can download the Genbank accession of GAPDH in this articel and know it.

  1. The authors need provide the images of single colony morphology of the representative isolates.

A: Images of single colony morphology of the representative isolates have shown in in my Master's thesis. I add a statement at lane 74.

Round 2

Reviewer 3 Report

The authors did not address the issue of materials and methods the reviewer has been pointed out. 

Author Response

First of all, thanks so much for your comments, we have revised the manuscript according to your comments. If you have any suggestions please let me known. Thanks.